# Early Mechanisms of Chemoresistance in Retinoblastoma

**DOI:** 10.3390/cancers14194966

**Published:** 2022-10-10

**Authors:** Michelle G. Zhang, Jeffim N. Kuznetsoff, Dawn A. Owens, Ryan A. Gallo, Karthik Kalahasty, Anthony M. Cruz, Stefan Kurtenbach, Zelia M. Correa, Daniel Pelaez, J. William Harbour

**Affiliations:** 1Bascom Palmer Eye Institute, University of Miami Miller School of Medicine, Miami, FL 33136, USA; 2Sylvester Comprehensive Cancer Center, University of Miami Miller School of Medicine, Miami, FL 33136, USA; 3Al-Rashid Orbital Vision Research Center, University of Miami Miller School of Medicine, Miami, FL 33136, USA; 4Interdisciplinary Stem Cell Institute, University of Miami Miller School of Medicine, Miami, FL 33136, USA; 5Simmons Comprehensive Cancer Center, UT Southwestern Medical Center, Dallas, TX 75390, USA; 6Department of Ophthalmology, UT Southwestern Medical Center, Dallas, TX 75390, USA

**Keywords:** retinoblastoma, chemoresistance, carboplatin

## Abstract

**Simple Summary:**

Despite advances in chemotherapy for retinoblastoma over the past three decades, chemoresistance remains a major source of ocular and systemic morbidity. Here, we studied the early molecular mechanisms leading to carboplatin resistance. Carboplatin is one of the most widely used agents in retinoblastoma, and it induced transcriptomic reprogramming involving the PI3K-AKT pathway, including the upregulation of ABC transporters and metabolic regulators. These findings nominate candidates for pharmacologic inhibition to circumvent chemoresistance and improve outcomes in retinoblastoma.

**Abstract:**

Retinoblastoma is the most common eye cancer in children and is fatal if left untreated. Over the past three decades, chemotherapy has become the mainstay of eye-sparing treatment. Nevertheless, chemoresistance continues to represent a major challenge leading to ocular and systemic toxicity, vision loss, and treatment failure. Unfortunately, the mechanisms leading to chemoresistance remain incompletely understood. Here, we engineered low-passage human retinoblastoma cells to study the early molecular mechanisms leading to resistance to carboplatin, one of the most widely used agents for treating retinoblastoma. Using single-cell next-generation RNA sequencing (scRNA-seq) and single-cell barcoding technologies, we found that carboplatin induced rapid transcriptomic reprogramming associated with the upregulation of PI3K-AKT pathway targets, including ABC transporters and metabolic regulators. Several of these targets are amenable to pharmacologic inhibition, which may reduce the emergence of chemoresistance. We provide evidence to support this hypothesis using a third-generation inhibitor of the ABCB1 transporter.

## 1. Introduction

Retinoblastoma is the most common eye cancer in children, accounting for approximately 11% of cancers in the first year of life [1,2]. Retinoblastoma is fatal if left untreated, but survival has increased dramatically over the past century due to improvements in diagnosis and treatment [3]. Currently, the most common eye-sparing treatment for retinoblastoma is chemotherapy, delivered intravenously or via ophthalmic artery infusion [3]. While chemotherapy has largely replaced radiation therapy and has reduced the need for enucleation, it continues to present major challenges related to ocular and systemic complications, severe vision loss, and treatment failure, owing to a narrow therapeutic index and chemoresistance [4,5,6]. Understanding the mechanisms of chemoresistance could allow for new pharmacologic strategies to expand the therapeutic window, improve treatment outcomes, and reduce complications. To date, most studies of chemoresistance in retinoblastoma have relied on highly passaged cell lines or surgical enucleation samples [7,8].

In this study, we employed newly derived low-passage retinoblastoma cells and next-generation sequencing methods to elucidate early molecular adaptations associated with repeated exposure to carboplatin, one of the most common therapeutic agents in retinoblastoma management.

## 2. Materials and Methods

### 2.1. Preparation of Cell Lines

Newly established low-passage RB028 cells were derived from tumor tissue retrieved at the time of primary enucleation by the senior author (J.W.H.) and cultured at 5% O_2_ in Dulbecco’s modified essential medium (DMEM)/F12 with 1X B-27 minus vitamin A (Life Technologies, Carlsbad, CA, USA), 1% penicillin-streptomycin, 2 mM Glutamax (Gibco, Waltham, MA, USA), 10 ng/mL basic fibroblast growth factor (bFGF) (PeproTech, Waltham, MA, USA, catalogue #100-18B), 10 ng/mL recombinant human stem cell factor (rSCF) (PeproTech, catalogue #300-07), and 20 ng/mL epidermal growth factor (EGF) (PeproTech, catalogue #AF-100-15). RB006 was established and maintained in a similar manner [9].

RB028 and RB006 were engineered to constitutively express GFP, which was cloned into pLenti-CMV-Puro-DEST backbone (Addgene, Watertown, MA, USA, Plasmid #17452). Briefly, 4 × 10^6^ H293T cells were plated onto 10 cm^2^ plates and transfected the following day with 5 μg of packaging vectors, 1.25 μg of pMD2.G (Addgene, Plasmid #12259), and 3.75 μg of psPAX2 (Addgene, Plasmid #12260). Viral supernatants were collected at 48 h and 72 h post-transfection, passed through 0.45 μm filters, and concentrated with polyethylene glycol. The concentrated viral mixture was then added to retinoblastoma cultures in 24-well plates and centrifuged for 1 h at 1000× *g* at 32 °C. After two days of incubation, cells were transferred to T25 flasks for recovery. After 1 week, cells were selected with 1 μg/mL of puromycin (Sigma, St. Louis, MO, USA, catalogue #5.08838.0001). GFP expression was confirmed by fluorescence imaging.

To generate chemoresistant RB028 cells, carboplatin was reconstituted in sterile water (10 mM carboplatin stock; MedChemExpress, catalogue #HY-17393) and further diluted for experiments using cell culture media. We treated cells with a range of carboplatin concentrations to identify the optimal dose for subtotal cell killing, and 1 μM was selected for further experiments. Subsequently, RB028 cells were treated with 1 μM carboplatin, which resulted in marked cell death after one week with scattered surviving cells. Fresh media was added as needed for the following 1–3 weeks until clusters of recovered cells were identified, at which time the cells were replated in media without carboplatin and grown to full recovery. This process was repeated for a total of 4 rounds of carboplatin treatment.

### 2.2. Cell Viability Analysis

Cells were mechanically dissociated with a P1000 pipette, and aggregates of approximately 500 cells/cluster were formed by plating into anti-adherence rinsing-solution-treated microwell plates (STEMCELL Technologies, Vancouver, BC, Canada, catalogue #34411 and #07010). Clusters were distributed using wide-bore tips in 96-well plates (~100 clusters per well) or 48-well plates (~300 clusters per well). Appropriate dilutions of carboplatin, tariquidar (Sellekchem, Zürich, Switzerland, catalogue #S8028), sodium dichloroacetate (DCA) (MedChemExpress, Monmouth Junction, NJ, USA, catalogue #HY-Y0445A), and BEZ-235 (MedChemExpress, catalogue #HY-50673) were added. Four images of each 96-well or 16 images of each 48-well were captured in the GFP-channel every 4 h over 8–10 days using the Incucyte Imaging System. Images were masked with a fixed GFP intensity threshold to remove the background signal. The total live area per image was summed for each well and used for downstream calculations. GraphPad Prism 9 was used for fitting nonlinear regression curves and identifying EC_50_.

### 2.3. RNA Sequencing

RNA was isolated from cell pellets using TRIzol (Ambion, Austin, TX, USA, catalogue #15596026). Libraries were prepared using the NEBNext Ultra kit and sequenced on Illumina NovaSeq (100 bp pair-end sequencing). Reads were trimmed using Trim Galore (DOI 10.5281/zenodo.5127899) and aligned to the human genome hg38 using STAR [10]. Gene counts were analyzed for differential expression using EdgeR [11]. Genes with a minimum count of less than 5 or a total minimum count of less than 10 were filtered out. GSEA analysis was performed using clusterProfiler [12] with preranked genes (FDR < 0.05, ranked by −log10[FDR]*sign[FC]). Platinum resistance genes were identified by overlapping differentially expressed genes with the platinum resistance database [13] using the filter “Up/down in Pt-resistant cells” as “UP”. Differentially expressed genes with FDR <0.05 and logFC >1 or <−1 were used, and gene ontology analysis was performed using Metascape.

### 2.4. Seeding Assay

Cells were mechanically dissociated with a P1000 pipette (pipetting 20–30 times) and then mixed 1:1.8 with Cultrex basement membrane hydrogel (Trevigen, Gaithersburg, MD, USA, catalogue 3433-010-01) and pipetted as 15 μL droplets in 6-well plates. This protocol was modified from the one previously described [14]. Droplets contained approximately 1000 cells and were incubated for 30 min at 37 °C for polymerization. After Cultrex gelation, media were added to cover cell droplets in each well. Calcein AM (Life Technologies, catalogue #L3224) was used to quantify final viability using the EVOS 9010 FL Auto Imaging System (Life Technologies). Total viable cluster counts were quantified with thresholding using ImageJ. The number of clusters per droplet were averaged across each well for statistical analysis using nonparametric t-tests in GraphPad Prism 9.

### 2.5. Barcoding and Single-Cell RNA Sequencing (scRNA-Seq)

CellTag barcoding plasmids containing constitutively expressed GFP construct were packaged into lentivirus with a titer of 2.5–3.5 × 10^8^ TU/mL [15,16]. A 200 μL aliquot of lentiviral Library #1 was added to low-passage naive RB028 cells in a 24-well plate, and the culture was centrifuged for 1 h at 1000× *g* in 32 °C. Cells were allowed to recover for at least one week. After a GFP signal was observed, cells were manually dissociated over ice with a P1000 pipette (pipetting 30–40 times) and passed through a 30 μm cell strainer. GFP+ RB028 cells were selected using fluorescence-activated cell sorting (FACS) with BD FACS SORP Aria-IIu, reaggregated using anti-adherence-rinsing-solution-treated microwell plates (STEMCELL Technologies, catalogue #34411 and #07010) and recovered overnight with rock inhibitor Y-27632 (10 μM; Selleckchem, catalogue #S1049). Cell clusters were lifted one week later and transferred into a flask for routine cell culturing protocol and then mechanically dissociated for each of the scRNA-seq experiments. In total, 10,000 cells were targeted for capture using 10X Illumina 3v3.1′ chemistry. The remaining cells were reaggregated into microwell plates with Y-27632 overnight. Once the culture recovered, cells were treated with 1 μM carboplatin to generate chemoresistance as described above.

### 2.6. scRNA-Seq Analysis

Cellranger (10× Genomics Cell Ranger 3.0.2) was used to align sequencing data to the hg38 reference genome modified to include CellTagUTR and eGFP [16,17]. DoubletFinder [18] was used for the identification and removal of doublets for downstream analyses. We prefiltered cells to exclude nFeature <200 and percent.mt >10. The doublet formation rate was estimated from quality-control images taken prior to sequencing (Sample 1 as 18.6%, Sample 2 as 22.4%, and Sample 3 as 20%). The cell IDs of the remaining 9266 cells after doublet removal were used for barcode detection and clonal analysis with the CellTagR pipeline [15,16]. Seurat v3 [19] was used for differential marker identification and visualization. The three samples were regressed for percent.mt and integrated with SCT. Clustering was performed with 30 dims and 0.4 resolution. Markers were identified using the Wilcoxon Rank Sum test, and GO analysis was performed with clusterProfiler [12,20] using filtered genes with adjusted *p* < 0.05. The percentage of cells in the 3×C (pct.1) and naïve (pct.2) groups in which the gene was detected was greater than 20%.

### 2.7. Quantitative PCR (qPCR)

Total RNA was extracted with TRIzol and reverse-transcribed using the iScript cDNA Synthesis Kit (Bio-Rad 1708890). qPCR was performed in triplicates using QuantStudio^®^ 7 RT-PCR System (Applied Biosystems) with the following parameters: 1 min of activation at 95 °C; 40 cycles of 10 s denaturation at 95 °C, and 60 s of annealing and extension at 60 °C. *RMB23*, *RNA18S*, and *SAP130* were used as internal controls. Primer sequences used are listed in Appendix A.

## 3. Results

### 3.1. Characterization of Carboplatin-Resistant Retinoblastoma Cells

RB028 cells that were naïve to chemotherapy (RB028^Naive^) or subjected to two cycles (RB028^2×C^) or four cycles (RB028^4×C^) of carboplatin treatment were challenged with 1 μM carboplatin for 7 days and then analyzed for cell viability (Figure 1A–C). Cells demonstrated increasing resistance to carboplatin in relation to their number of treatment/recovery cycles (EC_50_ for RB028^Naive^ ~3.15 μM [CI_95%_ = 2.399 to 4.007 μM]; EC_50_ for RB028^2×C^ ~11.96 μM [CI_95%_ = 10.57 to 13.54 μM]; EC_50_ for RB028^4×C^ ~22.73 μM [CI_95%_ = 18.01 to 30.47 μM]). This chemoresistance was found to be stable over as many as five passages (Appendix A). To simulate the formation of multicellular retinoblastoma seeds in the vitreous, which are associated with chemoresistance and poor outcome [21,22], we embedded mechanically dissociated cells into Cultrex droplets, allowed them to form into seed-like clusters, and measured their viability after 6 weeks (Figure 1D). Seed-like clusters formed by RB028^4×C^ cells exhibited greater viability without treatment and increased resistance to carboplatin compared to those formed by RB028^Naive^ cells (Figure 1E,F).

### 3.2. Transcriptomic Reprogramming of Carboplatin-Resistant Retinoblastoma Cells

Next, we wished to elucidate early transcriptomic events associated with the emergence of carboplatin resistance by performing RNA-seq in RB028^Naive^ and RB028^4×C^ cells. Using unsupervised principal component analysis and hierarchical clustering, RB028^4×C^ cells exhibited marked transcriptional changes compared to RB028^Naive^ cells (Figure 2A). Differential gene expression analysis revealed 1830 upregulated genes and 1992 downregulated genes in RB028^4×C^ cells compared to RB028^Naive^ cells (Appendix A and Figure 2B). Using gene set enrichment analysis, carboplatin resistance was associated with the downregulation of genes involved in oxidative phosphorylation, epigenetic gene regulation, and oxidative stress-induced senescence, and the upregulation of genes involved in PI3K/AKT signaling (Appendix A and Figure 2C).

### 3.3. Single-Cell Analysis Reveals Early Mechanisms of Carboplatin Resistance

There has been debate as to whether chemoresistance in retinoblastoma arises primarily by the selective expansion of preexisting resistant clones or by the generalized induction of resistance mechanisms [7] (Figure 3A). To perform lineage tracing of cells and to investigate potential clonal expansions, we uniquely labeled individual RB028^Naive^ cells using CellTag lentiviral barcoding libraries containing a GFP marker for sorting and then performed scRNA-seq before and after treatment with 1 μM carboplatin (Figure 3B). CellTags identified in 78% of RB028^Naive^ cells after doublet removal and low-quality filtering revealed diverse barcoding combinations (Appendix A). The reconstruction of lineage relationships generated from all three scRNA-seq samples resulted in 444 clones called from a total of 2649 cells (Figure 3C). The number of cells per clone ranged from 2 to 223 (mean 6.8 cells per clone, median 2 cells per clone). The visualization of all 9266 barcoded and non-barcoded cells with UMAP revealed nine cell clusters with no distinct separation based on carboplatin exposure (RB028^Naive^, RB028^2×C^, or RB028^3×C^) or clone size (Figure 3D). These results indicate that chemoresistance emerges through cellular reprogramming rather than the expansion of individual resistant clones. To gain insight into cellular adaptation resulting in resistance, we performed differential gene expression analysis. We find transcriptional reprogramming in RB028^3×C^ cells consistent with increased glycolysis and decreased oxidative phosphorylation (Appendix A and Figure 3E,F).

### 3.4. Pharmacologic Inhibition of ABCB1 Reverses Resistance to Carboplatin

Next, we wished to identify candidate genes that may contribute to carboplatin resistance and that may represent promising candidates for pharmacologic inhibition. We analyzed our RNA-seq dataset together with a curated database of over 900 genes that have been associated with platinum resistance [13]. For this analysis, we focused on genes that were upregulated by platinum therapy as the most likely targets for pharmacologic inhibition. Among 509 genes in the platinum resistance database and 1309 filtered genes in our RNA-seq dataset that were upregulated in association with platinum resistance, there were 40 genes in common (Figure 4A). This list remained highly enriched for genes involved in PI3K/AKT signaling pathway (adjusted *p* = 6 × 10^−7^) (Appendix A). Further, there were numerous genes in this list that are potentially druggable, including ABCB1, ABCC3, ABCC4, DKK3, EPHA2, ERBB4, and PDK4 (Figure 4B). The inhibition of PDK4 using DCA resulted in increased sensitivity to carboplatin (Appendix A). However, an even more pronounced effect was observed by inhibiting ABCB1, the most highly upregulated of these genes, in carboplatin-resistant retinoblastoma cells. The addition of the ABCB1 inhibitor tariquidar when treating RB028^4×C^ cells with 1 μM carboplatin resulted in a significant synergistic effect (Bliss score = 62.1%, observed inhibition = 77.2%, and excess over Bliss score = 15.1%) (Figure 4C–E). We further tested this combination in a second cell line derived from a chemotreated patient (RB006). As expected, RB006 was resistant to carboplatin compared to RB028^Naive^ but nevertheless became sensitized to carboplatin with the addition of tariquidar (Appendix A).

We then tested whether PI3K inhibition could reverse the carboplatin-induced expression of ABC transporters using BEZ-235, which has been shown to inhibit the PI3K/AKT pathway in retinoblastoma cells [23]. ABCB1 and ABCC3 were downregulated in RB028^4×C^ after treatment with 1 μM BEZ-235, both at baseline and in the presence of 1 μM carboplatin (Figure 4F). The addition of BEZ-235 chemosensitized RB028^4×C^ cells at low concentrations of carboplatin (Appendix A). The results for DCA and BEZ-235 were validated in RB006 as well (Appendix A). Interestingly, these results support a role for the PI3K/AKT pathway in carboplatin chemoresistance, yet phospho-AKT was not elevated in the RB028^4×C^ cells (Appendix A), suggesting stable epigenetic reprogramming.

## 4. Discussion

Despite progress in chemotherapy for retinoblastoma, ranging from intravenous to ophthalmic arterial infusion to most recently intravitreal approaches [24,25,26,27,28,29], ocular and systemic morbidity resulting from chemoresistance continue to pose a serious problem. Pharmacologic methods for circumventing chemoresistance are needed to reduce toxicity, increase efficacy, and improve patient outcomes. Unfortunately, the mechanisms of chemoresistance in retinoblastoma remain incompletely understood.

In this study, we focused on carboplatin because it is among the most widely used agents in retinoblastoma, and resistance to carboplatin continues to be a significant problem [13,24,27,30,31,32]. Carboplatin is a platinum compound that covalently binds DNA, resulting in DNA adducts and crosslinks that inhibit DNA replication and induce cell death [13,33,34]. Mechanisms of resistance to platinum compounds can include increased efflux of drugs, sequestration and detoxification of drugs, enhanced repair of DNA damage, reduced mismatch repair, the inhibition of apoptosis, adaptation to increased reactive oxygen species, increased autophagy, enhanced stress response, metabolic adaptation, transcriptional reprogramming, and other mechanisms [13]. We found that carboplatin induced extensive transcriptional reprogramming of retinoblastoma cells, particularly affecting pathways involved in metabolic adaptation. In particular, we observed a shift from pathways involved in oxidative phosphorylation to glycolysis, consistent with hypoxic adaptation and the Warburg effect, which have been linked to chemoresistance in retinoblastoma and other cancers [35,36,37,38,39].

To enrich for potential pharmacologic targets, we selected genes that overlapped between our RNA-seq dataset data and a curated database of genes associated with platinum resistance across multiple cancer types [13]. The resulting list was highly enriched for genes involved in the PI3K/AKT signaling pathway, which has been implicated in chemoresistance in retinoblastoma and other cancers [8,40,41]. PI3K/AKT signaling can promote chemoresistance by numerous mechanisms, including metabolic adaptation and the transcriptional activation of ABC transporters [40]. Indeed, the inhibition of PI3K with BEZ-235 resulted in the reversal of expression ABCB1 and ABCC3, which are multi-drug-resistant transporters that efflux a wide range of cytotoxic compounds [42]. ABCB1 has been associated with chemoresistance in retinoblastoma [7,43,44,45,46]. Cyclosporin has been used as a means of overcoming chemoresistance in retinoblastoma owing to its purported ability to inhibit ABCB1 [47], although this mechanism of action has been questioned [48]. Whereas cyclosporin was associated with substantial toxicity owing to its low potency as an ABCI1 inhibitor, there are now third-generation ABIB1 inhibitors such as tariquidar that demonstrate improved potency and lower toxicity [49]. To further validate ABCB1 as a potential mediator of carboplatin resistance, we treated retinoblastoma cells with carboplatin with or without tariquidar and found that this compound significantly reduced resistance to carboplatin.

A distinctive feature of this study was our use of recently developed technologies such as scRNA-seq and single-cell barcoding to dissect the early mechanisms of chemoresistance at the level of individual retinoblastoma cells. There has been controversy as to whether chemoresistance in retinoblastoma arises by the selective expansion of preexisting chemoresistant “cancer stem cells” or by generalized induction of resistance mechanisms [7,32,46]. Our findings from single-cell clonality analysis suggest that chemoresistance does not arise from a small number of resistant stem cells. Rather, many retinoblastoma cells appear to have the capacity for chemoresistance through transcriptomic reprogramming and metabolic adaptation, potentially through the PI3K/AKT signaling pathway. Although ABCB1 and other ABC transporters have been implicated as markers of “cancer stem cells” [50,51], they are also transcriptional targets of PI3K/AKT signaling [40], which is upregulated by carboplatin treatment. Thus, our findings suggest that the association of ABCB1 expression with chemoresistance in retinoblastoma is due to transcriptional reprogramming associated with PI3K/AKT signaling in many tumor cells, rather than outgrowth of chemoresistant stem cells.

## 5. Conclusions

Our findings are consistent with carboplatin resistance in retinoblastoma arising through the generalized transcriptomic reprogramming of tumor cells, rather than the selective outgrowth of a few preexisting cancer stem cells, which could have important implications for new therapeutic strategies. In addition to third-generation ABCB1 inhibitors, our findings nominate other clinically available compounds that may be inhibitors of chemoresistance, including PI3K inhibitors. Future work is needed to perform similar experiments using other chemotherapeutic agents used for retinoblastoma, to extend these findings to additional low-passage human retinoblastoma cells and to validate key findings in preclinical animal models.

## Figures and Tables

**Figure 1 cancers-14-04966-f001:**
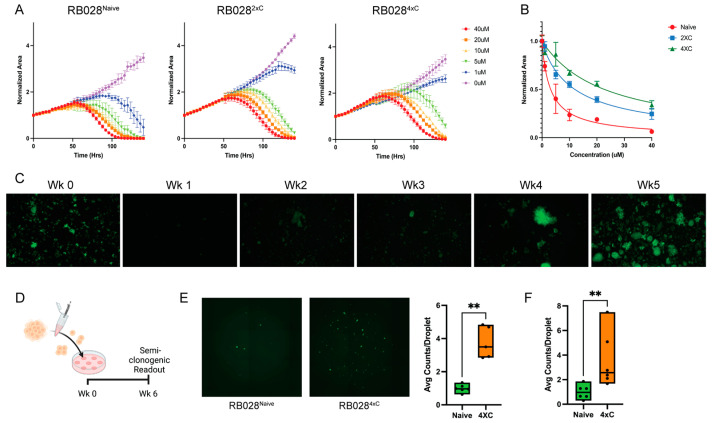
Sublethal carboplatin treatments induce chemoresistance in RB028^Naive^. (**A**) Normalized viability of RB028 cell lines challenged with carboplatin, monitored over 7 days. (**B**) Representative kill curve at Hour 100 for carboplatin challenge demonstrates gain of chemoresistance (EC_50_ for RB028^Naive^ ~3.15 μM; EC_50_ for RB028^2×C^ ~11.96 μM; EC_50_ for RB028^4×C^ ~22.73 μM). (**C**) Recovery of RB028^Naive^ treated with 1 μM carboplatin over five weeks. (**D**) Schematic of semi-clonogenic assay. (**E**) Representative images of RB028^Naive^ and RB028^4×C^ droplets at Week 6. Semi-clonogenic assay demonstrates higher average number of viable clusters per droplet for RB028^4×C^ at baseline and (**F**) with 5 uM carboplatin (** *p* < 0.01).

**Figure 2 cancers-14-04966-f002:**
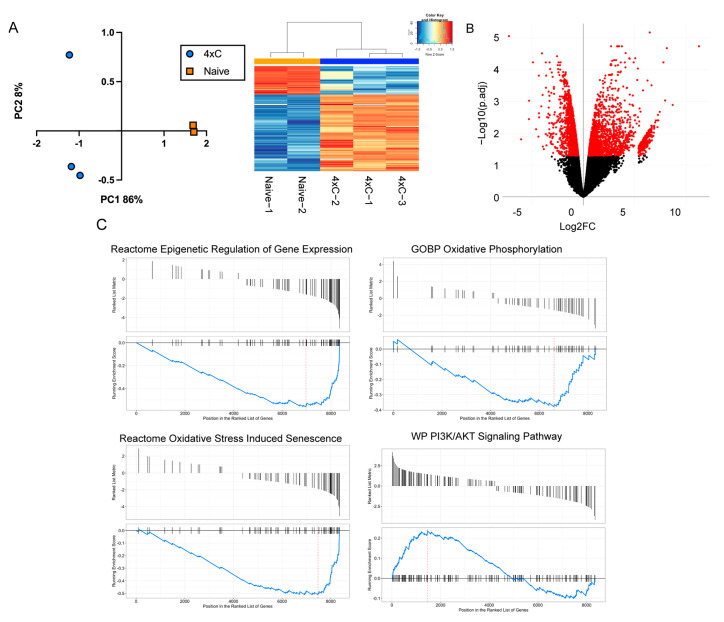
Transcriptomic changes associated with recurrent carboplatin exposure. (**A**) PCA plot and heatmap comparing RB028^4×C^ versus RB028^Naïve^. (**B**) Volcano plot of RB028^4×C^ versus RB028^Naïve^ (red dots represent genes with FDR < 0.05). (**C**) GSEA analysis identifies pathways involved in early resistance reprogramming events (adjusted *p* < 0.05).

**Figure 3 cancers-14-04966-f003:**
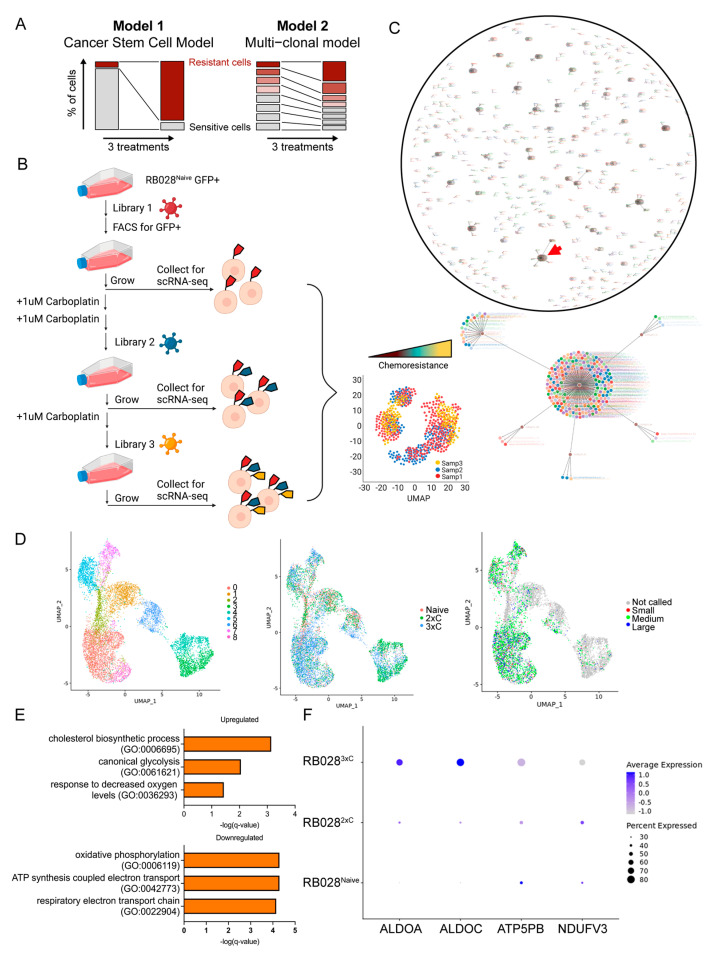
Clonal analysis with barcoded scRNAseq. (**A**) Simplified models of chemoresistance development in retinoblastoma. (**B**) Schematic of clonal analysis experiment. RB028^Naive^ cells barcoded with CellTag Library #1 were enriched by fluorescence-activated cell sorting (FACS). RB028^Naive^ were treated twice with 1 μM carboplatin and transduced with CellTag Library #2 to generate RB028^2×C^, which were then treated once more with 1 μM carboplatin and transduced with CellTag Library #3 to generate RB028^3×C^. In total, 10,000 cells of RB028^Naive^, RB028^2×C^, and RB028^3×C^ were captured for scRNAseq analysis. (**C**) Reconstruction of lineages between scRNAseq samples with force-directed graphing. Each point represents an individual cell. Each line represents clonal relationships between cells. A total of 2649 cells were assigned to 444 clones. The largest clone with 223 cells (red arrow) is shown enlarged. (**D**) UMAP plot depicting assigned Seurat clusters (left), sample of origin (middle), and clone size (right). (**E**) Gene ontology pathways derived from genes differentially expressed in RB028^3×C^ compared to RB028^Naive^. (**F**) Dot plot of select differentially expressed genes associated with glycolysis and oxidative phosphorylation across RB028^Naive^, RB028^2×C,^ or RB028^3×C^ cells.

**Figure 4 cancers-14-04966-f004:**
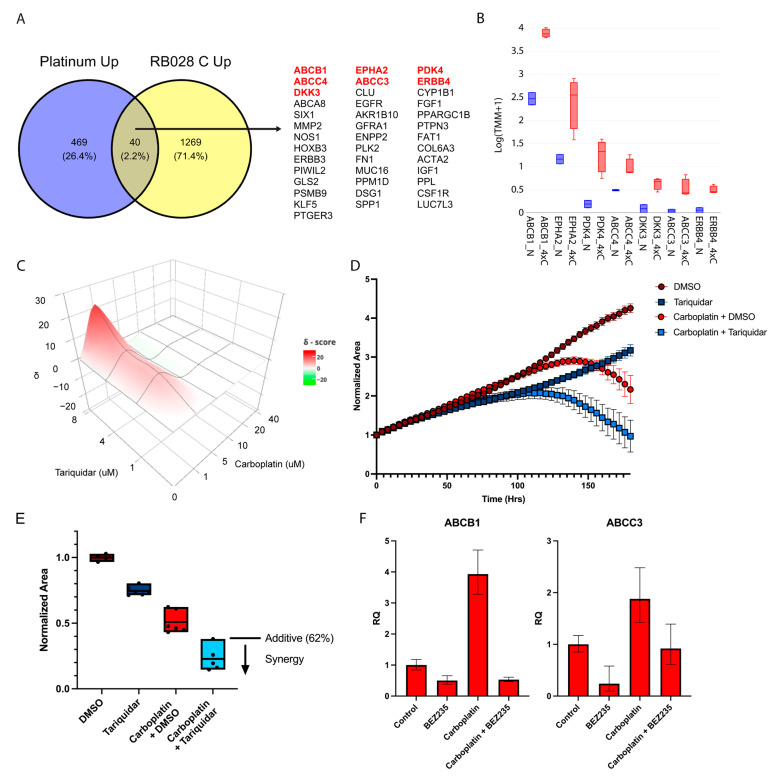
Carboplatin resistance potentiated by ABCB1 inhibitor tariquidar. (**A**) Venn diagram of genes upregulated in a platinum resistance database (Platinum Up) and genes upregulated in RB028^4×C^ compared to RB028^Naive^ cells (RB028 C Up). Genes analyzed further in panel B are indicated in red. (**B**) RNA-seq data of select genes upregulated by carboplatin in RB028^4×C^ (red) and RB028^Naive^ (blue) cells. (**C**) Representative 3D synergy plot of RB028^4×C^ using Bliss scores. (**D**) Normalized viability of RB028^4×C^ cells monitored over 8 days after challenge with carboplatin (1 μM) along with tariquidar (4 μM). (**E**) Normalized viability of RB028^4×C^ cells at the endpoint from panel C for the indicated treatment groups (Bliss score = 62.1%, observed inhibition = 77.2%, and excess over Bliss score = 15.1%). (**F**) qPCR of ABCB1 and ABCC3 with BEZ-235 (1 μM) and carboplatin (1 μM) on Day 3. Error bars represent confidence intervals set to 95th percentile.

## Data Availability

RNA-seq and scRNA-seq data from RB cell lines have been deposited in dbGaP (accession number pending).

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
