# Peer review of "Early Mechanisms of Chemoresistance in Retinoblastoma"

_cancers, 2022, doi:10.3390/cancers14194966_

Round 1

Reviewer 1 Report (Previous Reviewer 1)

Authors have addressed the concerns.

Author Response

NA

Reviewer 2 Report (Previous Reviewer 2)

These are interesting and significant findings directly pertinent to the human disease and therapy. The manuscript is improved with new experiments and revision.  I support its publication after minor edits.

If the new gene expression data in Fig. 4C is significant (right and left) it should be noted in the legend and figure.

I agree, altered epigenetics could activate PI3K signaling as noted by the authors. If Akt is not the key downstream target of newly active PI3K, then what is? Also, the key PI3K player might not be PI3KCA, since BEZ235 inhibits each of them.

Author Response

This manuscript is a resubmission of an earlier submission. The following is a list of the peer review reports and author responses from that submission.

Round 1

Reviewer 1 Report

The observations that the resistance arise from reprogramming are interesting. However, authors need to address the comments below.

1. One of the weaknesses is that the studies were carried out with cell lines derived from one parental line.

2. Authors need to comment on EMT. Do the early mechanism also involve the EMT genes? PDK4 was shown to induce EMT.

3. Increased PDK4 would explain the metabolic changes. Does knockdown of PDK4 in the resistant cells increase sensitivity?

4. Fig. 4: The conclusion of "synergy" is not convincing - It appears (from panel D) to be an additive effect.

5. Line 153, authors need to include the resistance data over 5 passages as a supplemental file.

Reviewer 2 Report

The authors conduct an important study on how primary retinoblastoma cells acquire resistance to carboplatin, a DNA damaging agent and standard of care treatment for this cancer. They generate resistant cells, conduct single-cell transcriptomic profiling to identify differences and pick up key candidate genes. Several ABC-transporters become overexpressed and then efflux carboplatin to elicit resistance. Next, co-administering a novel ABC-transporter inhibitor along with carboplatin caused synergistic cell death. Bar-coding technology is used to demonstrate that resistant RB clones come from a variety of sources, not a single resistant progenitor cell. Thus, the conclusions could be directly applicable to the human disease and may provide a novel therapeutic angle for aggressive cases.

However, several issues exist with the manuscript’s current state that should be addressed before publication.

Major Points 

Authors claim in all sections that reprogramming in response to excess carboplatin primarily involves the PI3K-AKT signaling pathway, but I don’t see where or how this claim is supported by primary data. GSEA data in Figure 2C generated based on RNA-seq data doesn’t mention PI3K. Neither does the GO definitions shown in Figure 3D. On P6,L218 the authors state that PI3K stats were “adjusted p=6x10-7” but I don’t see the source. Although references to PI3K can be found in supplemental Table S1 (GSEA), these data are not directly shown in the paper. This would seem to be necessary to support the claim that the response is “primarily” mediated through PI3K.

To support this claim, the authors should present related GSEA in the manuscript and could conduct several straightforward experiments to begin to validate it. First, active Akt-P should be detectable by immunoblotting or IHC as elevated in RB0284xC versus Rb028Naive. Second, as per the experiment in Figure 4C, combined PI3K inhibition and carboplatin administration should reduce viability of RB cells. Third, as a potential mechanism connecting these pathways, the authors could easily test if PI3K inhibition reverses the carboplatin acquired expression of some of the ABC transporter genes by qPCR in the RB0284xC line. Otherwise, "reliance" on PI3K/AKT for this phenotype should be toned down.

Last, were any statistics used to establish the conclusion of “selective outgrowth of a few preexisting cancer stem cells”? Authors should describe more extensively in the manuscript and Figure S1 how their conclusion was determined, and perhaps more precisely how the data should look different if most cells instead arose from a single clone.

Minor Points

·         Figure 1B. No apparent statistics are used to demonstrate significance of this key sub-figure.

·         Text in Figure 3 is particularly illegible and a higher resolution image should be used.

·         Schema in Figure 3A is quite small and the corresponding legend text don’t describe the procedure (although it is described in detail in Methods). It seems too succinct here for the complicated nature of the procedure and more details might help.
